# Aberrant Positions of the Chemosensory Neurons in the Neurotransmitter-Release Mutant *unc-13*

**DOI:** 10.3390/ijms252312956

**Published:** 2024-12-02

**Authors:** Eduard Bokman, Ido Padro Kalij, Alon Zaslaver

**Affiliations:** Department of Genetics, Silberman Institute of Life Science, Edmond J. Safra Campus, The Hebrew University of Jerusalem, Jerusalem 9112001, Israel

**Keywords:** *C. elegans*, *unc-13*, sensory neurons, neuroanatomy, synaptic activity

## Abstract

Secretion of neurotransmitter- and neuropeptide-containing vesicles is a regulated process orchestrated by multiple proteins. Of these, mutants, defective in the *unc-13* and *unc-31* genes, responsible for neurotransmitter and neuropeptide release, respectively, are routinely used to elucidate neural and circuitry functions. While these mutants result in severe functional deficits, their neuroanatomy is assumed to be intact. Here, using *C. elegans* as the model animal system, we find that the head sensory neurons show aberrant positional layout in neurotransmitter (*unc-13*), but not in neuropeptide (*unc-31*), release mutants. This finding suggests that synaptic activity may be important for proper cell migration during neurodevelopment and warrants considering possible anatomical defects when using *unc-13* neurotransmitter release mutants.

## 1. Introduction

Animal genetic models offer powerful means to study gene-to-function relationships. One such model, the *Caenorhabditis elegans* (*C. elegans*) nematode, is widely used to elucidate functional roles of individual genes and neurons [1,2]. Moreover, owing to its fully-mapped 302-neuron connectome [3,4,5] and its compatibility with a myriad of genetic manipulations, *C. elegans* is instrumental in elucidating neuron-specific functions as well as network-wide principles that underlie various behavioral outputs [6,7]. In that respect, two mutants, the neurotransmitter (*unc-13*) and the neuropeptide (*unc-31*) release mutants continuously play pivotal roles [8,9,10]. *Unc-13* is a highly conserved gene involved in docking the neurotransmitter-containing vesicles to the synaptic release sites, and subsequently priming them to be fusion-competent with the plasma membrane [11,12,13]. Indeed, studies in worms, flies, and mice demonstrated that mutants, defective in *unc-13*, are impaired in neurotransmitter release [8,14,15]. This impaired vesicle release was shown to affect transmission but not the layout and the general anatomy of the nervous system. For example, while *unc-13* defective *C. elegans* animals show severe uncoordinated movement, the architecture of the GABAergic motor neurons as well as their synaptic composition is intact [8]. Likewise, normal synaptic structures were observed in flies and mouse hippocampal neurons [14,15].

The *unc-31* gene codes for an essential component for the Ca^2+^-dependent exocytosis of dense-core vesicles [10,16,17]. These dense-core vesicles contain neuropeptides and monoamines that modulate presynaptic or postsynaptic functions via binding and activating G-protein coupled receptors. While both *unc-13* and *unc-31* participate in Ca^2+^-dependent exocytosis processes, in the *C. elegans* nervous system, they have parallel roles: *Unc-31* acts in dense-core vesicles while *unc-13* acts in synaptic vesicles.

In *C. elegans*, the head chemosensory system consists of 12 bilaterally symmetric pairs of neurons [6]. These neurons are located within the head amphids, which are laterally located sensilla that are open to the outside at the circumference of the worm’s mouth. Each neuron has a bipolar structure: anteriorly, it extends a neurite to the tip of the animal’s nose, while the other branch is an axon-like extension that forms in- and out-going synapses with other neurons within a dense nerve ring. These neurons sense chemicals in the environment and play prominent roles in many physiological and behavioral outputs, including chemotaxis, mechanosensation, osmotaxis, thermotaxis, and more [6,7,18,19].

Herein, we analyzed the anatomical position of the amphid chemosensory neurons in both *unc-13* and *unc-31* mutants. We find that while the anatomical layout of the neuropeptide release mutant *unc-31* is indistinguishable from that of wild-type (WT) animals, the cell bodies in the *unc-13* mutants are disarranged. This finding challenges the currently-held assumptions that *unc-13* mutants are impaired in vesicle release only. Moreover, it suggests that intact neurotransmitter release may play important roles in shaping and organizing the anatomical layout of a nervous system.

## 2. Results and Discussion

To study the organizational layout of the amphid chemosensory neurons, we imaged a transgenic strain in which all amphid neurons express a florescent marker (Figure 1A–C). For imaging, we used a confocal microscope capturing z-stack images at a resolution of 0.6 µm apart, sufficient for cellular resolution detection of individual neurons. Using our previously developed image analysis pipeline we segmented and identified individual neurons based on the nuclear expression of mCherry (see Methods). Notably, mCherry nuclear localization is crucial as it provides the cellular resolution required for the accurate identification of individual neurons. The result of this anatomical analysis generates the 3D organization of the cell’s nuclei (Figure 1D).

We next compared the neural spatial organization of WT animals to that of *unc-31* and *unc-13* mutant strains (Figure 1D–F). The neural layout of the *unc-31* mutants resembled that of WT animals, where the neurons within each of the two amphid halves were positioned in parallel. In contrast, the layout of the *unc-13* mutants showed a stark difference where the two amphid halves were tilted such that they intersected at the dorsal part.

To quantitate this anatomical difference, and to provide a statistical measure for the significance of the aberrant organization, we measured the angle formed between the two bilateral sides of the amphid neurons. First, we constructed best-fit planes for the right and the left halves of the amphid (Figure 1D–F). Each plane crosses the neurons positioned on one lateral side such that the summed distances of the nuclei from the plane are minimal (Methods). Indeed, these planes are almost in a perfect parallel position in WT and *unc-31* mutant animals, and the average angle between the planes is 15.51 ± 14.18 degrees and 18.50 ± 19.18, respectively (Figure 1D,F,G). In contrast, in the *unc-13* mutant strain, the two halves of the amphid intersect one another at the dorsal side forming a significantly larger angle between the two planes of 29.09 ± 16.66.

A complementing approach to quantify neural displacement observed in the *unc-13* mutant is to calculate the distance between the corresponding bilateral right and the left types of neurons. We find that while the average distance between individual bilateral neurons is similar in WT and *unc-31* mutant animals, this distance is significantly higher in *unc-13* mutants (Figure 1H). Specifically, the increased distance is observed between the internal neurons positioned along the dorso–ventral axis (namely, AWC, AWA, ASH, ASE, and ADF), whereas the distance between the five dorsal bilateral neurons (ADL, ASK and ASI, AWB, and ASG) and the most ventral bilateral neurons (ASJ) is intact and resembles the distance found in WT animals. To provide a quantitative measure for the greater distancing in the *unc-13* mutant, we calculated the regression line denoting the distance between the bilateral neurons in the anterior–posterior axis (Figure 1H, see methods). We found that the slope of the *unc-13* mutant is two-fold steeper than that of the WT, a statistically significant difference which was confirmed using a bootstrap analysis (*p* < 0.001, Figure 1I). Together, these results indicate that the abnormal neural position is mainly due to the greater distance between the bilateral neurons positioned in the inner part of the anterior–posterior axis of the animal.

**Figure 1 ijms-25-12956-f001:**
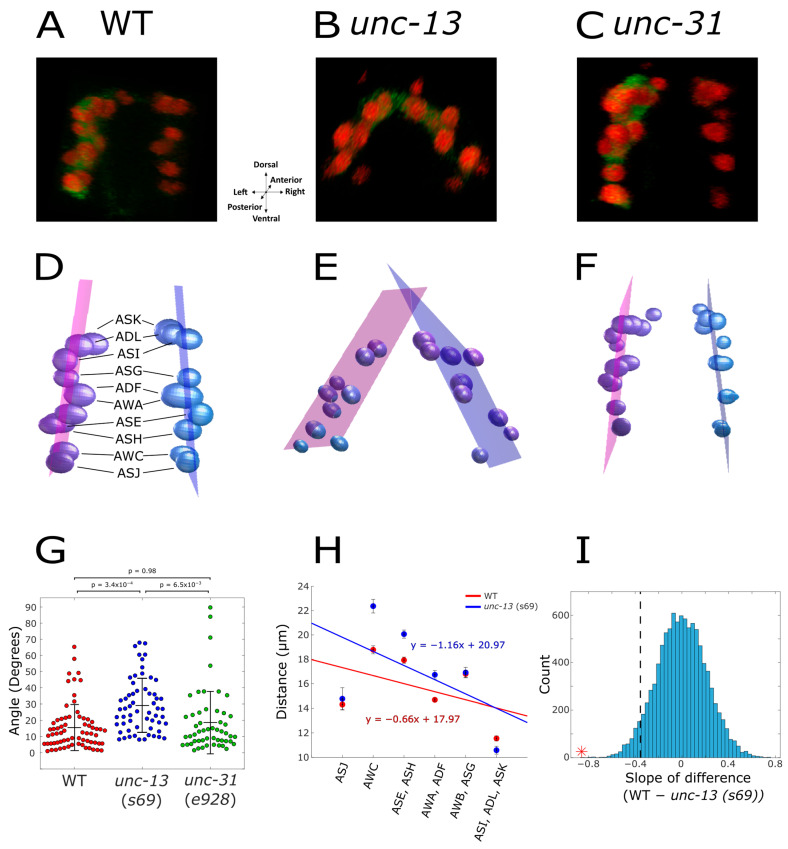
**The anatomical layout of the chemosensory neurons is aberrant in *unc-13*, but not in *unc31*, mutant animals**. (**A**–**C**) 3D imaging of the amphid chemosensory neurons in (**A**) WT, (**B**) *unc-13*, and (**C**) *unc-31* mutant animals. This image is produced by stacking together the individual Z-plane images. The imaged strain expresses GCaMP in the cytoplasm and mCherry in the nuclei of all amphid neurons (*osm-6*::GCaMP, *osm-6*::mCherry-NLS [20]). (**D**–**F**) Spatial localization of the nuclei of the amphid chemosensory neurons in (**D**) WT, (**E**) *unc-13*, and (**F**) *unc-31* animals. An in-house image analysis pipeline segments, identifies, and labels the individual chemosensory neurons from both lateral sides of the amphid [20,21]. A best-fit plane encompassing all the neurons on each side of the amphid was constructed. (**G**–**I**) The angle that two planes of the sensory neurons form at the dorsal side of the amphid (WT n = 70, *unc-13* n = 59, *unc-31* n = 53) (**G**), and the distance between the corresponding two bilateral neurons together with the regression lines (WT, red; *unc-13*, blue, panel **H**). (**I**) A bootstrap analysis which calculates the slope for randomly chosen WT and *unc-13* neurons indicated that the observed difference between the strains is statistically significant (*p* < 0.001, see methods). The dashed line represents the 5% percentile of the bootstrap estimated slope difference. The red asterisk denotes the observed slope difference. These measurements are from worms inserted into a microfluidic device.

The imaging procedure detailed above was performed on animals restrained in a microfluidic device [22]. This approach allows for measuring accurate neural position by imaging multiple planes while the animals are held in place. However, inserting the worms into a tight restraining channel squeezes and pressurizes them. Consequently, it could be that the observed neural displacement of *unc-13* mutants is due to reduced resistance to external mechanical forces, rather than neuroanatomical-related defects.

To distinguish between the two possibilities, we reimaged the WT together with two alleles of mutant *unc-13* (*s69* and *e51*) without restraining them in a microfluidic device. For this, worms were anesthetized and laid on an agar pad, such that they did not experience any external force (Methods). Comparing the angle between the two bilateral planes showed a significantly larger angle in both of the *unc-13* mutants (Figure 2A), thus, corroborating the findings using the microfluidic chips. In fact, while the angle of WT animals was very similar (16.84° ± 7.84), the angle of the *unc-13* mutant was larger than the one observed when using microfluidic devices (42.71° ± 15.79 (*S69*), and 38.02° ± 9.10 (*e51*) vs. 29.09° ± 16.66 (*S69* restrained)). Thus, the difference in the angles is even more pronounced when not squeezing the animals inside the narrow microfluidic channels. Measuring the distance between the bilateral neurons showed that in both *unc-13* mutants, the sensory neurons in the middle of the anterior–posterior axis are further apart than the neurons at the edge of this axis, similar to what was observed when imaging the neurons in the microfluidic device (Figure 2B,D). Likewise, the slope of the regression line between the neurons was two-to-three-fold steeper than in the WT (*s69*: *p* = 0.0036, *e51*: *p* = 0.0133, Figure 2B–D).

Altogether, here we show that animals, defective in neurotransmitter release (*unc-13*), but not neuropeptide release (*unc-31*), have aberrant positioning of the chemosensory neurons’ cell bodies. This neural disposition is evident by the greater distance between the left and the right bilateral sensory neurons. Our observation is particularly surprising, considering that *unc-13* mutants are generally smaller than WT worms [23]. Cell migration in the chemosensory system has been implicated in both developmental and adult neurogenesis processes [24], and impaired cell migration during development is associated with abnormalities in olfactory bulb morphology [25,26]. While the role of adult neurogenesis in olfactory sensing remains disputed, the proper creation and migration of new olfactory neurons is required for some learning and memory processes [27]. Our findings raise the interesting possibility that intact synaptic activity may be necessary for proper cell migration during the development of the nervous system.

In *C. elegans*, synaptic structures are organized in clusters, presumably to support compartmentalized local computations along the neurites, a feature that may enhance the computational capacity of a neural network [28]. While the synaptic organization of the amphid neurons was not studied herein, previous studies in *C. elegans*, using the same *unc-13* mutant allele (*s69*), found no anatomical changes in the GABAergic neuromuscular junctions, where the distribution of the postsynaptic GABA receptors was indistinguishable from that of WT animals [8]. Similarly, while the priming and secretion of neurotransmitter vesicles were severely impaired, normal synaptic structures were observed in mutant flies, as well as in mouse hippocampal neurons [14,15]. Thus, while dysfunctional synaptic activity may not affect proper synaptic formation, it may affect neurodevelopmental-related migratory processes of the cell bodies.

## 3. Materials and Methods

### 3.1. Strains

ZAS280 *In*[*osm-6*::GCaMP3, *osm-6*::ceNLS-mCherry-2xSV40NLS] [20].

ZAS325 is a cross between ZAS280 and *unc-31*(*e928*) [21].

ZAS371 is a cross between ZAS280 and *unc-13*(*s69*).

ZAS529 is a cross between ZAS280 and *unc-13*(*e51*).

### 3.2. Worm Cultivation

All strains were grown on NGM plates pre-seeded with OP 50 and kept at 20 °C. Age synchronization was performed by bleaching. All experiments were done on young-adult animals, 3 days post-bleaching.

### 3.3. Imaging the Chemosensory System

For microfluidic-based imaging, individual worms were inserted into a microfluidic chamber [22] where they were partially anesthetized with 10 mM levamisole and left to habituate for 10 min [20,21,29,30,31,32]. For the agar pad experiments, a smallchunk of NGM agar was placed onto a slide. A ~15 µL drop of 10 mM levamisole was pipetted onto the agar, and small groups of 5–8 worms were picked into the drop, gently covered with a coverslip, and left to habituate for 10 min.

Imaging was performed on a Nikon A1R+ confocal laser scanning microscope with water immersion ×40 (1.15NA) objective, and *Z*-axis intervals of 0.5–0.8 µm. The system was controlled by the Nikon NIS-elements software (v5.02.01).

### 3.4. Neuron Identification

Neural nuclei were segmented based on the nuclear mCherry signal using an algorithm developed by [33]. Neuronal identities were determined visually based on anatomical positions and relative size and expression levels as in previous studies [20,21,29]. Briefly, the dorsal triplet of ASK, ADL, and ASI is identified first, as these neurons are usually positioned in a straight line. Then, ASH is identified as the largest and strongest expressing neuron on the ventral side, and ASJ, ASE, and AWC are identified by their positions relative to ASH, with ASJ being ventro-posterior, AWC being ventro-anterior, and ASE being posterior to ASH. Finally, the middle four are disambiguated. ASG and ADF have the strongest expression on the posterior and anterior sides, respectively. AWA is larger and more medial, while AWB is smaller and more lateral. Neurons that could not be unambiguously identified were removed from the analysis.

### 3.5. Statistical Analysis

The angles between the two sides of the amphid were determined by fitting a plane to the 3-dimensional positions of the nuclei centroids on each side, and calculating the angle at which the planes intersect. Worms with fewer than 4 identified neuron pairs were excluded. The *p*-values were obtained using a one-way ANOVA.

For the distances within the bilateral neuron pairs, the neuron classes were assigned ranks based on their typical relative position along the dorso–ventral axis. Neurons that are close together on this axis were given the same rank. From ventral to dorsal the ranks are as follows: ASJ = 1, AWC = 2, (ASE, ASH) = 3, (AWA, ADF) = 4, (AWB, ASG) = 5, (ASI, ADL, ASK) = 6. Ranking along the dorso–ventral axis was used because of the difficulty of establishing the objective anatomical dorso–ventral axis for precise distance calculations, as well as to simplify the re-sampling in the permutation analysis. Distances between the left and right sides of each neuron pair were calculated, and a regression line was fit to the mean distances of the ranks. The *p*-values were obtained using a permutation test. All distances within each rank were shuffled and randomly assigned to WT or mutant. The mean mutant distance per rank was then subtracted from the mean WT distance, and a regression line was fit to the result. The observed value of the slope of this regression line was compared to the slopes of 10,000 random permutations.

## Figures and Tables

**Figure 2 ijms-25-12956-f002:**
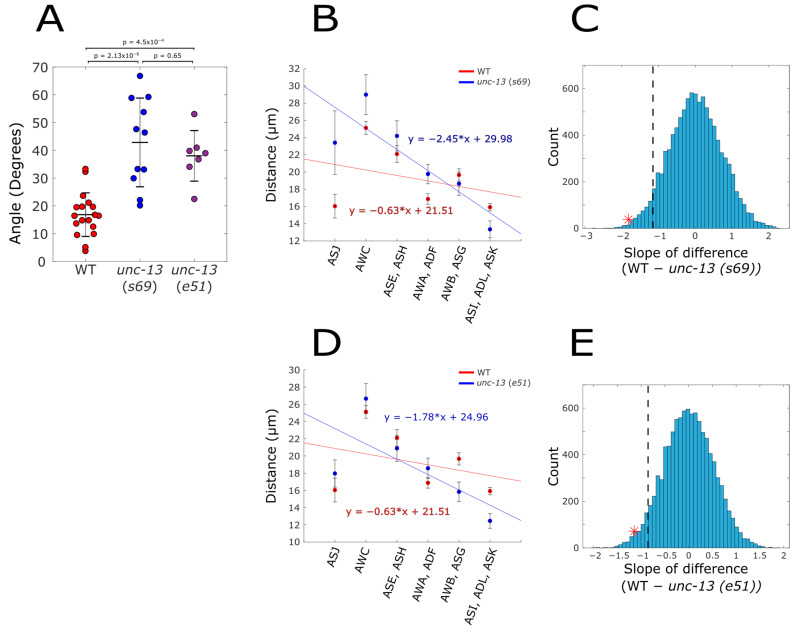
**Aberrant neural position of *unc-13* mutants is also observed in unrestrained animals.** Animals were placed on an agar pad and imaged (WT n = 18, *unc-13* (*s69*) n = 11, *unc-13* (*e51*) n = 7). (**A**) The angle that the two planes of the sensory neurons form at the dorsal side of the amphid. (**B**,**D**) The distance between the corresponding two bilateral neurons together with the regression lines (WT, red; *unc-13*, blue). (**C**,**E**) Bootstrap analysis (as detailed in Figure 1I) to indicate the significance. The dashed line represents the 5% percentile of the bootstrap estimated slope difference. The red asterisk denotes the observed slope difference. (**C**) *p* = 0.0036, (**E**) *p* = 0.0133.

## Data Availability

The raw images of the results are available upon request.

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
