# Peer review of "Aberrant Positions of the Chemosensory Neurons in the Neurotransmitter-Release Mutant unc-13"

_ijms, 2024, doi:10.3390/ijms252312956_

Round 1
Reviewer 1 Report
Comments and Suggestions for Authors
In the manuscript titled “Aberrant positions of the chemosensory neurons in the neurotransmitter- release mutant unc-13”, authors demonstrated that the head sensory neurons show aberrant positional layout in neurotransmitter (unc-13), but not in neuropeptide (unc-31), release mutants using C. elegans as the model animal system, which presents a very interesting phenomenon. However, this report merely describes the anatomical aberrations in an impaired synaptic activity mutant, subsequent studies underlying mechanisms are required.
I am not qualified to assess the image processing program and analysis method used in this article.
Here several minor questions remain to be addressed.
1. In Results and Discussion, P6, Comparing the angle between the two bilateral planes showed a significantly larger angle in the unc-13 mutants (Fig 3J), thus ------. where is Fig 3J?
2. In Materials and Methods, P8, Neural nuclei were segmented based on the nuclear mCherry signal using an algorithm developed by (Toyoshima et al., 2016). Does the sentence after “developed by” needs parentheses? and pay attention to the writing format.
3. In Materials and Methods, P8, How neuronal identities were determined by visual observation based on anatomical positions, what method is it exactly? the author should provide detailed identification methods that were used in the process.
Reviewer 2 Report
Comments and Suggestions for Authors
This study is a brief communication describing anatomical differences in the placement of chemosensory neurons in unc-13 mutant C. elegans. It is concise, well written and presents innovative analyses. However, further information is needed to rule out artifact and clarify importance.
1. Please provide quantification of worm measurements. Could the reported differences be due to overall abnormal proportions?
2. Please define all abbreviations.
3. Please clearly state the n for each experimental group.
4. Please speculate on the functional consequence of these anatomical differences. How would this contribute to the phenotype of the unc-13 mutant?
5. Please avoid general, non-specific descriptions. For example in the sentence: "While both unc-13 and unc-31 participate in Ca-dependent exocytosis processes, in the C. elegans nervous system, they have parallel roles: Unc-31 acts in dense-core vesicles while unc-13 acts in synaptic vesicles." What specifically do unc-13 and unc-31 do? i.e., do they facilitate vesicle docking or fusion? What do the mutant specifically do? What are the mutant phenotypes?
Round 2
Reviewer 2 Report
Comments and Suggestions for Authors
The authors have addressed all of my concerns. Very nice and concise presentation of data indicating the potential importance of Unc-13 for neuronal migration. This descriptive study lays the foundation for future mechanistic studies.